# Alpha-Gal Bound Aptamer and Vancomycin Synergistically Reduce *Staphylococcus aureus* Infection In Vivo

**DOI:** 10.3390/microorganisms11071776

**Published:** 2023-07-08

**Authors:** Matthew K. Doherty, Claire Shaw, Leslie Woods, Bart C. Weimer

**Affiliations:** 1Population Health and Reproduction, University of California Davis, Davis, CA 95616, USA; mkdoherty@ucdavis.edu (M.K.D.); clashaw@ucdavis.edu (C.S.); 2California Animal Health and Food Safety Laboratory, University of California Davis, Davis, CA 95616, USA; lwwoods@ucdavis.edu

**Keywords:** antimicrobial, MRSA, aptamer antibiotic, antimicrobial resistance, sepsis

## Abstract

Methicillin-resistant *Staphylococcus aureus* (MRSA) is a pervasive and persistent threat that requires the development of novel therapies or adjuvants for existing ones. Aptamers, small single-stranded oligonucleotides that form 3D structures and can bind to target molecules, provide one possible therapeutic route, especially when presented in combination with current antibiotic applications. BALB/c α-1, 3-galactosyltransferase (−/−) knockout (GTKO) mice were infected with MRSA via tail vein IV and subsequently treated with the αSA31 aptamer (n = 4), vancomycin (n = 12), or αSA31 plus vancomycin (n = 12), with split doses in the morning and evening. The heart, lungs, liver, spleen, and kidneys were harvested upon necropsy for histological and qPCR analysis. All mice treated with αSA31 alone died, whereas 5/12 mice treated with vancomycin alone and 7/12 mice treated with vancomycin plus αSA31 survived the course of the experiment. The treatment of MRSA-infected mice with Vancomycin and an adjuvant aptamer αSA31 reduced disease persistence and dispersion as compared to treatment with either vancomycin SA31 alone, indicating the combination of antibiotic and specifically targeted αSA31 aptamer could be a novel way to control MRSA infection. The data further indicate that aptamers may serve as a potential therapeutic option for other emerging antibiotic resistant pathogens.

## 1. Introduction

*Staphylococcus aureus* is a leading cause of bacteremia in the United States with 100,000 cases a year, although this pathogenic bacteria is also asymptomatically carried by 20–30% of the human population [1]. Pathogenic colonization with *S. aureus* can manifest in a variety of ways, including skin and soft tissue infections (SSTI), bone, joint, and implant infections, and pneumonia and septicemia [2]. *S. aureus* infection can also lead to chronic carriage in wounds and the onset of various toxicoses like toxic shock syndrome [3]. *S. aureus* infection and carriage is not limited to humans. Many different species of animals experience comparable diseases, such as bovine mastitis, and zoonotic transmission from pigs to humans has been observed [4,5]. Given the clinical relevance and prevalence of *S. aureus* in both human and animal populations, it is concerning that multi-drug-resistant *S. aureus* is on the rise globally [6].

In the 1950s, penicillin was successfully used to control *S. aureus* infections, but such widespread use led to the appearance of penicillin-resistant *S. aureus* only a few short years after the introduction of the groundbreaking antibiotic [6]. Following a similar trajectory, methicillin-resistant *S. aureus* (MRSA) was observed in 1961, only two years after clinics switched from using penicillin to methicillin to control *S. aureus* [6]. Now, more than 63% of all *S. aureus* isolates are resistant to at least one antibiotic, with many being resistant to more than four antibiotics [7]. More concerningly, resistance to daptomycin and vancomycin, antibiotics considered the last resort in treating MRSA infections, has been reported [8,9]. While MRSA infections are primarily associated with nosocomial transmission, community-acquired infections are becoming increasingly common [10]. With the rapid development of resistance and increasing modes of transmission, it is clear that novel antibiotics or adjuvants are needed to combat MRSA.

The development of antibiotic–adjuvants, compounds that increase the efficacy of antibiotics, could make existing therapies more effective at lower doses and with fewer side effects. One example is the commonly prescribed antibiotic–adjuvant mixture Augmentin, which combines the β-lactam antibiotic amoxicillin with the β-lactamase inhibitor clavulanic acid. The use of Augmentin, as opposed to amoxicillin alone, restores activity against amoxicillin-resistant bacteria and increases bacterial clearance [11,12,13]. Novel adjuvants are being investigated [14,15], but considering the rapid rate of genome evolution and the continuing emergence of antibiotic resistant strains, it is imperative that alternate strategies are also investigated [7,16].

One such alternate strategy is to leverage the patient’s own immune system and preexisting antibodies (Abs). Abs modulate the immune response and are naturally produced against foreign antigens, like the gal-α1,3-gal (α-gal) epitope [17]. α-gal is a carbohydrate structure not produced by humans, Old World monkeys, or apes, but it is commonly found across many other mammals and bacteria [18]. Diet and the gut microbiota are thought to be the primary source of α-gal exposure, and given the consistent contact with both, there is likely continuous antigenic stimulation in the host [19,20,21,22,23,24]. This consistent α-gal exposure in humans leads to high levels of anti-α-gal Abs, comprising up to 10% of total circulating Abs and up to 1% of total IgG in humans [19,20,25]. In an evolutionary context, anti-α-gal Abs may have emerged as Old World monkeys faced viruses, taking advantage of the hosts own α-gal surface modification [26]. Thus, simultaneously losing the ability to produce α-gal epitopes and creating Abs capable of flagging these invaders became an advantageous way to fend off viruses [26]. In addition to virus flagging, anti-α-gal Abs are active against galactosyltransferase-positive (GalT^+^) bacteria via activation of the complement pathway [27].

Anti-α-gal Abs readily react with α-gal on the surface of viruses, bacteria, and animal tissues consumed via diet. In a hypothetical mechanism like that of a sandwich ELISA, if the α-gal sugar were bound to a target molecule with high affinity to a pathogen (a.k.a. the primary Ab), the bound pathogen would then display α-gal from its surface. This would flag the pathogen for binding by circulating anti-α-gal Abs (a.k.a. the secondary Ab) and signal the adaptive immune system to clear the infection before mounting a new Ab response against specific pathogen-associated molecular patterns (PAMPs). Targeting molecules, the primary Ab in the ELISA metaphor, have already been suggested and studied for pathogen binding. Perdomo et al. [28] showed that HIV-1 was neutralized via α-gal-linked CD4-mimetic peptides in serum, suggesting that small molecules or peptides could be used as agents for pathogen-specific targeting.

Another class of possible targeting molecules are aptamers, single-stranded oligonucleotides (ssDNA or ssRNA) that assume three-dimensional conformations capable of binding small molecules, proteins, or whole cells. Aptamers specifically able to bind α-gal have earned the moniker alphamer, or α-mer. Aptamers are similar to Abs in that they often bind specific targets, analogous to Abs’ binding of epitopes. However, aptamers have multiple advantages over Abs including a smaller size, low immunogenic potential, higher target affinity and selectivity, and easy synthesis in vitro, and they allow for modifications that change their stability and specificity [29]. The successful application of α-mers in vitro to control group A *Streptococcus* bacteria has already been reported [30]. Aptamer modification to resist nuclease degradation is commonly achieved by altering the phosphate backbone with sulfur to create a phosphothioate bond at a variety of locations in the sequence. Additionally, 3′ and 5′ capping with other molecules offers increased stability in serum [29]. The current modifications and the previously listed advantages allow for the potential use of aptamers in lieu of more complex Abs applications in therapeutic and diagnostic contexts [29,31].

As one possible path for combatting antibiotic resistance, the continued development of new aptamers increases the likelihood that successful target molecules could be found in tandem with the emergence of new resistant pathogens. A high-throughput selection process for novel aptamers has already been developed, known as the Systematic Evolution of Ligands by Exponential Enrichment (SELEX) [29], and can be readily applied to seek out novel therapeutic solutions. Aptamers for the specific binding of whole *S. aureus* cells have already been reported by Cao et al. [32], establishing that the production of aptamers targeting entire bacteria, not just molecular fragments, is possible. Further, Cao et al. demonstrated active binding with a high affinity and selectivity for *S. aureus* [32].

This present study builds on the aptamer study by Cao et al. [32] study and proposes that an aptamer from their study, SA31, coupled to α-gal (αSA31) would rescue α-1, 3-galactosyltransferase (−/−) knockout (GTKO) mice from induced MRSA sepsis. In this study, we showed that treatment with both vancomycin and αSA31 rescued more mice and resulted in lower bacterial loads in multiple organs than treatment with vancomycin or αSA31 alone. Together, these data suggest that α-mers may be an effective antibiotic adjuvant to reduce bacterial distribution and persistence during the antibiotic treatment of sepsis.

## 2. Materials and Methods

### 2.1. Drugs

All aptamers and α-mers were obtained from BioSearch (Novato, CA, USA). Aptamers were prepared from lyophilized powder in 0.9% sterile saline. Vancomycin (Hospira, Lake Forest, IL, USA) was obtained from the UC Davis School of Veterinary Medicine, prepared as per manufacturer’s protocol, and diluted to 20 mg/mL in 0.9% sterile saline prior to use. The GpC SA31 oligonucleotide was 5′capped with AminoC6 and produced by Integrated DNA Technologies Inc., Coralville, IA, USA.

### 2.2. Bacterial Strains and Growth Conditions

The methicillin-resistant strain, *S. aureus* ATCC 33591, was used in this study. *S. aureus* ATCC 33591 was thawed from −80 °C vials, transferred twice into BHI broth at 37 °C and grown to late log phase before use. The minimum inhibitory concentration of vancomycin for this strain was determined to be between 3 and 6 µg/mL, based on the growth inhibition of *S. aureus* ATCC 33591 cultured overnight in BHI with 0–15 µg/mL vancomycin.

### 2.3. In Vitro Cell Culture

Colonic epithelial cells (Caco-2; ATCC HTB-37) were obtained from the American Type Culture Collection (Manassas, VA, USA) and grown as per the manufacturer’s instructions in T-25 flasks. Subsequently, for compound treatment, cells were seeded to a density of 10^5^ cells/cm^2^ in a 96-well plate using DMEM/High Modified (Thermo Scientific, Rockford, IL, USA) with 16.6% fetal bovine serum (FBS) (HyClone Laboratories, Logan, UT, USA), non-essential amino acids (Thermo Scientific), 10 mM MOPS (Sigma, St. Louis, MO, USA), 10 mM TES (Sigma), 15 mM HEPES (Sigma) and 2 mM NaH_2_PO_4_ (Sigma). Cells were incubated at 37 °C in 5% CO_2_ for 14 days post confluence to allow for differentiation to occur prior to adhesion assays [33]. Human and mouse macrophage cell lines (THP-1 and RAW264.7, respectively) were obtained from the American Type Culture Collection (Manassas, VA, USA) and grown as per the manufacturer’s instructions in T-75 or T-125 flasks. For phagocytosis assays, cells were seeded to a density of 10^5^ cells/cm^2^ in 48-well plates [34,35].

### 2.4. Aptamer Stability

A single aptamer (SA31) was used for this study based on previous binding studies as reported by Cao et al. [32]. Different versions of SA31 were used to determine the influence of individual modifications on serum stability. End-capped (5′-α-gal-SA31-NH_2_-3′ and 5′-α-gal-SA31-3C-3′) uncapped (5′-NH2-SA31-OH-3′) forms of SA31 were incubated in serum from *Homo sapiens* and *Mus musculus,* or in the presence or absence of RQ1 DNase 1 (1 unit/100 µL) (Promega, Madison, WI, USA), Exonuclease1 (2 units/100 µL), or T5 Exonuclease (20 units/100 µL) (New England Biolabs, Ipswich, MA, USA) at 37 °C for 0–24 h. Degradation was measured using qPCR using primers previously reported by Cao et al. [32]; amplification conditions were changed to 95 °C for 3 min, 40 cycles of 95 °C for 10 s, and 52 °C for 30 s. The PCR reaction mix was composed of iQ Sybrgreen 2X Master Mix (Bio-Rad, Hercules, CA, USA) forward and reverse primers at s 100 nM final concentration, nuclease free H_2_O (Gibco, Grand Island, NY, USA), and template DNA from samples at a final volume of 25uL per reaction well. All qPCR was performed using the Bio-Rad CFX96 real-time PCR instrument (Bio-Rad, Hercules, CA, USA). Amplified products were verified using melt curve analysis from 50 °C to 95 °C with a transition rate of 0.2 °C/s and band size verification using the Agilent BioAnalyzer 2100 (Agilent, Santa Clara, CA, USA) with the RNA chip to measure ssDNA products. Half-life was calculated using the equation t_1/2_ = (elapsed time × log 2)/log (initial [drug]/final [drug]).

### 2.5. Bacterial Adhesion Assay

*S. aureus* ATCC 33591 was grown as described above and re-suspended to an OD_600_ of 0.2 in DMEM/highly modified medium containing non-essential amino acids, 10 mM MOPS, 10 mM TES, 15 mM HEPES, and 2 mM NaH_2_PO_4_ without FBS before use in the adhesion assay. The differentiated epithelial cells were washed once with 200 µL of PBS just prior to the addition of the bacteria. The bacterial suspension (50 µL) was mixed with appropriate amounts of aptamers and mixed by inversion for 15 min prior to addition to the differentiated Caco2 cells at a final multiplicity of infection of 1:1000. The Caco2 cells treated with bacteria and aptamer treatments were incubated at 37 °C in an atmosphere containing 5% CO_2_ for 60 min to allow for the bacteria to associate with the epithelial cells. After incubation, each treatment was aspirated and the Caco2 monolayer washed three times with 200 µL of Tyrodes buffer (pH 7.2) [36,37] to remove non-adhered bacterial cells. Adhered bacterial concentration was determined as described by Elsinghorst et al. [38] and others [34,35], except qPCR was used to determine the bacterial count using 50 µL of a commercial lysis buffer (AES CHEMUNEX, Inc., Cranbury, NJ, USA) that lyses mammalian and bacterial cells prior to qPCR. The amplification parameters for mammalian GAPDH primers (forward: ACCACAGTCCATGCCATCAC; reverse: TCCACCACCCTGTTGCTGTA) was 95 °C for 5 min, followed by 40 cycles at 95 °C for 15 s, 56 °C for 30 s, and 72 °C for 30 s, then a final extension at 72 °C for 1 min. Bacterial detection was carried out in using qPCR for the *S. aureus femA* gene [39] with the PCR conditions as described by Nadkarni et al. [40] using universal 16s primers.

### 2.6. S. aureus Growth Inhibition with α-SA31

To determine if αSA31 inhibited the growth of *S. aureus,* a growth curve was conducted with αSA31 in an automated plate reader (Beckman Coulter DTX 800, Brea, CA, USA) by measuring A_600_ every hour for 24 h in triplicate. Log-phase *S. aureus* was adjusted to an OD_600 nm_ = 0.1 in a 96-well plate with BHI broth and 20% GTKO mouse serum. αSA31 was added to the appropriate wells at 5 ng/µL and incubated at 37 °C for 24 h. The plate was mixed for 30 s before each OD measurement.

### 2.7. Phagocytosis Assay

*S. aureus* ATCC 33591was grown as described above to stationary phase and re-suspended to an OD_600_ of 2. The bacterial suspension was incubated with or without 20% serum, with or without 150 ng/µL aptamer, and finally with or without α-galactose-1,3 monoclonal IgM (Enzo Life Sciences, Inc., Farmingdale, NY, USA) diluted to 1:50, separately for 15 min each, prior to resuspension in FBS free RPMI/DMEM and addition to macrophages at a final multiplicity of infection of 1:100. *S. aureus* cells were washed 3X with PBS between incubations. The cells treated with pretreated *S. aureus* were incubated at 37 °C in an atmosphere containing 5% CO_2_ for 60 min to allow for the bacteria to associate with the cells. After incubation, each treatment was aspirated and the cells washed three times with 200 µL of PBS to remove non-adhered/phagocytized bacterial cells. Cells used to analyze phagocytosis were then incubated for 2 h in FBS-free media containing 100 µg/mL gentamycin. This media was removed, the cells washed once with PBS, and then mammalian and bacterial cells were lysed prior to qPCR using 50 µL of a commercial lysis buffer (AES CHEMUNEX, Inc., Paris, France). CFU MRSA per cell was determined using qPCR to calculate copies *S. aureus femA* per copies mammalian GAPDH. The amplification parameters for the mammalian GAPDH and *S. aureus femA* primers were optimized for single-plate amplification to 95 °C for 5 min, followed by 40 cycles at 95 °C for 15 s, 62 °C for 30 s, and 72 °C for 30 s, then a final extension at 72 °C for 10. Phagocytized CFU/cell were subtracted from total associated CFU/cell to determine adhered cell amounts.

### 2.8. Ethics Statement

This study was carried out in accordance with the recommendations in the Guide for the Care and Use of Laboratory Animals of the National Institutes of Health. All efforts were made to minimize animal suffering. All animal protocols received prior approval by the UC Davis Institutional Animal Care and Use Committee (Protocol Number: 16284).

### 2.9. Animals

BALB/c GT^−/−^ (GTKO) mice were bred and maintained under standard conditions by the Center for Laboratory Animal Science at the University of California, Davis, which is accredited by the American Association for Accreditation of Laboratory Animal Care. Each mouse was immunized using rabbit ghost erythrocytes prepared as described by Dodge et al. [41] before use in the study. Briefly, 50 mL Alsevers Rabbit Blood (bioMérieux, Inc., Durham, NC, USA) was centrifuged at 1000× *g* for 20 min and washed three times with cold (4 °C) 1X PBS. The washed pellet was resuspended in 10–12 mL of cold 1X PBS. A total of 3 mL of the suspension was aliquoted into 27 mL of cold 1X PBS, incubated at 4 °C for 30 min, and centrifuged at 40,000× *g* for 20 min. The pellet was resuspended in 10 mL of cold sterile distilled deionized water, centrifuged at 20,000× *g*, 4 °C for 20 min, and washed with cold sterile distilled deionized water until the supernatant turned light pink to clear. The pellet was resuspended with cold 1X PBS and stored at 4 °C. Protein contents were quantified via the NanoDrop 2000 (Thermo Scientific, Waltham, MA, USA). Each mouse was immunized with 0.1 mL of approximately 1 mg/mL rabbit ghost erythrocyte suspension, boosted 1 week later, and monitored by α-gal Ab ELISA.

Blood was collected daily from each mouse (~50 µL) via tail nick. Terminal blood draws were collected via intracardiac puncture immediately after euthanasia via CO_2_ overdose at the end of treatment or after loss of 20% of body weight. All blood samples were transferred into BD Microtainer Serum Separator tubes (Becton Dickinson and Company, Franklin Lakes, NJ, USA), and the serum separated as per manufacturer’s protocol. The serum was transferred into a clean 0.5 mL microcentrifuge tube and stored at −80 °C for further analysis. At necropsy, the kidneys, lungs, and spleen were collected from each mouse via dissection. Approximately one half of each organ was immediately placed in 10% buffered formalin acetate at room temperature for histology while the remaining organ tissue was placed in 1.5 mL micro-centrifuge tube and temporarily incubated on ice prior to storage at −80 °C for later analysis.

Histology was performed and scored by a single pathologist (L. Woods) blinded to the treatment groups. The scores were averaged for bacteria, inflammation, and necrosis (B/I/N) on a scale of 0–4, with 0 being healthy tissue with no bacteria and 4 being severely damaged tissue or numerous bacteria.

Bacterial detection was carried out in each tissue using qPCR for the *S. aureus femA* gene [39] with the PCR conditions as described by Nadkarni et al. [40] using universal 16S primers. The tissue samples were homogenized using 3 mm glass beads in 2 mL minibeadbeater tubes in volumetric equivalents of sterile PBS with a Minibeadbeater (BioSpec Products, Inc., Bartlesville, OK, USA) for 30 s. Homogenates were further diluted for a total organ dilution of 1:100 into commercial lysis buffer (AES CHEMUNEX, Inc.) for total lysis according to manufacturer’s protocol. Tissues were diluted to 1:1000, in total, for the determination of MRSA colony-forming units (CFU/mg) organ after qPCR.

### 2.10. In Vivo Accumulation of αSA31

The in vivo concentration of αSA31 was determined after injecting mice intravenously (IV) twice daily with αSA31 concentrations ranging from 150 µg/kg/day to 10,000 µg/kg/day. Serum was collected daily prior to morning dosage and qPCR, as described above, was used to determine the concentration of αSA31 in the serum of each animal.

### 2.11. α-gal Ab ELISA

Black 96-well plates (Thermo Scientific) were coated overnight at 4 °C in a 100 mM bicarbonate/carbonate buffer (pH 9.6) with 50 µL of 2.5 mg/mL gal-α1,3-gal-human serum albumin with a 14-atom spacer (V-Labs Inc., Covington, LA, USA). The wells were rinsed twice with phosphate-buffered saline plus 0.2% Tween 20 (pH 7.2) (PBST) and blocked with 200 µL of Superblock (Thermo Pierce, Thermo Scientific, Waltham, MA, USA) overnight at 4 °C. Before use, the plates were washed twice with PBST, prior to the addition of 50 µL mouse serum was serially diluted in PBST and added to each well for 15–20 min at 37 °C while rocking; then, plates were washed twice with PBST. The secondary antibody, goat anti-mouse IgGAM conjugated to fluorescein isothiocyanate (FITC) (Invitrogen, Waltham, MA, USA) was diluted in PBS with goat serum, as per manufacturer’s protocol, and 50 µL was added to each well. Plates were sealed with foil and incubated for 20 min at 37 °C while rocking, washed once with PBST and washed once with PBS prior to the addition of 50 uL of PBS for fluorescence detection using a Beckman Coulter DTX 800 (Beckman Coulter, Inc., Indianapolis, IN, USA) with an excitation wavelength of 495 nm and an emission wavelength of 535 nm. Background fluorescence from a blank well was subtracted to report ELISA response. All assays were carried out in triplicate.

### 2.12. Statistics

Statistical analyses were performed using Prism 5 for Mac (GraphPad Software Inc., La Jolla, CA, USA) and JMP10 for Mac (SAS Institute Inc. Cary, NC, USA). Serum αSA31 concentrations were analyzed via a repeated measures 2-way ANOVA and a Bonferroni multiple comparisons post-test. Anti-α-gal ELISA statistics were performed using 1-way ANOVA and a Bonferroni multiple comparisons post-test. Adhesion assay statistics were performed using 1-way ANOVA and a Dunnett’s multiple comparisons test to no drug control. Phagocytosis assay statistics were performed using a 1-way ANOVA and Tukey’s HSD post-test. Histology scores and aptamer half-life statistics were performed using a 2-way ANOVA and Tukey’s HSD post-test. Organ MRSA load statistics were segregated by survival and performed using nonparametric tests and Tukey’s HSD post-test or Student’s *t*-tests.

## 3. Results

### 3.1. Alphamer Characterization

αSA31 activity was initially characterized by examining in vitro the inhibition of MRSA growth and cellular adhesion. The addition of αSA31 to cell culture medium resulted in a significant decrease in the (*p* < 0.05) association of MRSA in vitro, while a control anti-flu aptamer [42] had no impact on adherence, even with increasing concentrations (Appendix A). αSA31 treatment actively blocked MRSA adhesion to Caco2 cells by approximately 50%, a reduction that remained constant despite the increasing dosage. αSA31’s ability to inhibit MRSA growth was also examined. No difference was observed in the growth of MRSA incubated in the presence of 20% mouse serum, either with or without 12.5 ng/µL of αSA31 (Appendix A), demonstrating that αSA31 is neither bactericidal nor bacteriostatic.

To verify that the α-gal moiety and SA31 remained linked while under the in vivo conditions of this study, the αSA31 combined aptamer was incubated for 60 min total in phosphate–citrate buffer at acidic, neutral, and basic conditions. Separation of the moiety and aptamer was evaluated by Bioanalyzer at incubation intervals of 15 min. Two distinct bands, at the appropriate location for αSA31 and unbound SA31, were observed in all pH conditions (Appendix A). Under acidic conditions, more SA31 was present than the combined αSA31. Conversely, both the neutral and the basic conditions had more αSA31 than unbound SA31, indicating that the α-gal SA31 linkage should remain intact in blood, though a small amount of unbound SA31 will likely be present. The results of the in silico analysis of the predicted secondary structure of the aptamer (Appendix A) was comparable to the results of the structural analysis conducted by Cao et al. [43].

### 3.2. 5′-Capping Did Not Affect Stability

The stability of 5′-α-gal-SA31-NH_2_-3′ (αSA31NH_2_), 5′-α-gal-SA31-3C-3′ (αSA31) and 5′-NH2-SA31-OH-3′ (SA31) was tested via endonuclease and exonuclease treatment (Table 1). In nuclease-free water alone, αSA31NH_2_ and αSA31 were significantly more stable (*p* < 0.001) than SA31. No degradation of either αSA31NH_2_ or αSA31 was observed after 24 h at 37 °C. The addition of endonuclease or either exonuclease did not lead to any significant differences in degradation times between the three aptamers. T5 exonuclease was added to test if the aptamer modifications protected them against 5′ exonuclease attack. Protection against 5′ exonuclease attack was not conferred by either the 5′ α-gal or the amine linkage. In human and mouse serum, the three aptamers displayed different patterns of stability. While αSA31NH_2_ was significantly more stable (*p* < 0.01) in human serum, no significant difference was observed between the stability of αSA31NH_2_ and αSA31 in mouse serum.

### 3.3. αSA31 Increased Phagocytosis In Vitro

αSA31 was assessed for its ability to increase the phagocytosis of MRSA in vitro. Mouse macrophage cells were added to MRSA that had been pretreated with serum, serum and SA31, serum and αSA31, or PBS only (Figure 1A). MRSA pretreated with GTKO mouse serum alone significantly increased (*p* < 0.0003) the adhesion of MRSA cells to mouse macrophages when compared to MRSA pretreated with PBS. The pretreatment of MRSA with serum and SA31 showed that a similar number of MRSA cells were phagocytized, as seen with the serum-only pretreatment. Notably, over 3.5-fold more (*p* < 0.05) pretreated MRSA were phagocytized when also treated with αSA31, compared to all other treatments. Similar experiments were performed using human serum and human macrophages, but no significant difference was observed between the treatments (Figure 1B)

The αSA31 aptamer contains three known CpG motifs (Appendix A). Given the CpG motif is a known PAMP, the role of these motifs in αSA31 on phagocytosis was assessed. A control aptamer was constructed that switched the three CpG motifs to GpC, creating an aptamer without the CpG PAMPs. Both SA31 and the modified GpC SA31 were assayed in the presence or absence of 10 µM chloroquine to determine TLR9 receptor involvement, which is a previously described interaction [44]. No significant difference was observed between SA31 and GpC SA31, regardless of treatment with chloroquine (Appendix A), indicating no involvement of CpGs and TLR9 in MRSA phagocytosis.

To confirm that anti-α-gal antibodies recognize αSA31 in vitro, MRSA pretreated with αSA31 was incubated in the presence or absence of a monoclonal anti-α-gal IgM [45]; then, following incubation, mouse macrophages were added and phagocytotic activity was evaluated. The phagocytosis of MRSA incubated with αSA31 and anti-α-gal IgM was blocked by over 10-fold compared to MRSA tagged with αSA31 alone (Figure 2). The anti-α-gal Ab was able to detect the α-gal moiety on the α-mer in vitro, so αSA31 was selected for testing in the in vivo studies.

### 3.4. In Vivo Alphamer Accumulation

In vivo accumulation and potential toxicity of αSA31 assessed prior to performing the infection rescue studies. Mice were dosed in both the morning and evening with αSA31 at total doses ranging from 300 to 10,000 µg/kg/day. Serum concentrations were then determined for each animal and no adverse symptoms were observed at any tested dose. In all mice, an immediate spike in serum αSA31 concentration was seen on day 1; however, the αSA31 did not show any accumulation over time (Figure 3A). The serum levels of αSA31 decreased over the treatment period despite daily administration. Mice treated with the maximum dose of 10,000 µg/kg/day αSA31 had significantly higher (*p* < 0.01) serum concentrations on days 2 and 3, compared to day 1, and showed a stabilization at approximately 2 ng/mL serum. The serum concentration of αSA31 in mice with the 10,000 µg/kg/day dose showed a marked decrease at day 4. To examine a possible mechanism of α-mer instability and to explain the observed serum concentration decrease, the level of α-gal antibody, anti-α-gal Abs, was also assessed in the serum before the first dose and after the first day. Circulating anti-α-gal was significantly lower at both dosing concentrations following α-mer treatment (*p* < 0.0008) (Figure 3B).

### 3.5. αSA31 Treatment during MRSA Sepsis

The in vivo antimicrobial activity of αSA31 was tested via the application of αSA31 at increasing doses in combination with vancomycin to septic mice. Sepsis was induced with a 1 × 10^9^ CFU/mouse dose of MRSA, as determined by preliminary experiments and prior published studies [46,47,48,49]. Mice were intravenously infected with

MRSA and treated with up to 10,000 µg/kg/day of αSA31, with no addition of vancomycin. All mice with induced sepsis and αSA31 only treatment did not survive beyond day 2 (Appendix A). Septic mice given Vancomycin (60 mg/kg/day) or Vancomycin plus 10,000 µg/kg/day αSA31 aptamer were partially rescued (Figure 4A). A total of 5/12 mice treated with vancomycin alone and 7/12 mice treated with the antibiotic–aptamer combination survived the course of the experiment.

The in vivo stability of αSA31 was compared between infected and uninfected mice to determine the effect of an active infection on aptamer circulation. Mice infected with MRSA and treated with the vancomycin-αSA31 combination had in vivo αSA31 stability like that seen in uninfected mice on day 1 (Appendix A). The serum levels of αSA31 began to drop between day 2 and day 3 in infected mice compared to uninfected mice. The serum concentrations between the two groups were significantly different on day 3 (*p* < 0.05). Vancomycin treatment of uninfected mice should have no effect in vivo on αSA31 stability.

### 3.6. αSA31 Treatment Plus Vancomycin Resulted in Lower MRSA Organ Loads

The bacterial load (CFU/mg organ) in the kidney, lung, heart, liver, and spleen (Figure 4B–F, respectively) from each mouse was determined using qPCR and further evaluated via histological examination. The distribution of bacteria through the observed organs differed by survival status and by treatment (Appendix A). Amongst mice that died due to infection, aptamer treatment at 2400 or 300 µg/kg/day significantly reduced the bacterial load in the lungs as compared to untreated animals (*p* ≤ 0.05). Mice treated with vancomycin that died had a significantly higher amount of MRSA in the lungs (*p* < 0.05) and significantly lower amount of MRSA in the spleen (*p* < 0.02) compared to mice treated with vancomycin plus 10,000 µg/kg/day αSA31 (Appendix A). Mice treated with vancomycin plus αSA31 that survived the experiment had less MRSA in the heart (*p* = 0.05) and significantly lower bacterial loads in the lung, liver, and kidney (*p* = 0.02, *p* ≤ 0.05, and *p* < 0.0004, respectively) compared to mice treated with vancomycin alone. Spleens from the group treated with Vancomycin and αSA31 had three-fold more MRSA than the mice treated with vancomycin alone.

### 3.7. Histology

Animals treated with vancomycin plus αSA31 showed significantly less, by a two-way ANOVA (*p* < 0.02), bacterial foci than animals treated with vancomycin alone. The difference was especially pronounced in the kidney. MRSA distribution across the organs was also significantly different (*p* < 0.0008) between vancomycin + αSA31 and vancomycin-only animals. Though no significant difference in necrosis was detected between vancomycin + αSA31 and vancomycin treatment groups, there was a significant difference (*p* < 0.001) in the distribution of necrosis amongst the organs within each treatment group (Appendix A).

## 4. Discussion

MRSA infection is a persistent threat, even more so now that the array of effective antibiotic therapies is diminishing. Additionally, the transition from primarily nosocomial transmission to increasing community-acquired infections only elevates the concern around developing effective MRSA treatments [7]. Novel therapies that are effective against MRSA are needed, especially those that result in fewer side effects than current antibiotic methods. Anti-α-gal Abs, which humans naturally produce en masse, is one option in the fight against MRSA. Preformed anti-α-gal Abs could be redirected to recognize MRSA through a bound α-gal aptamer, thereby flagging the immune system to begin clearing the infection. An aptamer that readily binds to whole *S. aureus* was identified in a previous study [50] and applied here in vivo to control MRSA infection in mice. To increase this SA31 aptamer’s stability and to make it immunogenic, and thus a target of anti-α-gal Abs, SA31 was modified with a 5′ α-gal and a 3′ three carbon cap.

Previous work from Cao et al. [32] showed that SA31 modified with fluorescent probe FITC binds to *S. aureus* specifically, but other 5′ or 3′ alterations could impact the binding of SA31 to MRSA. The adhesion assay in this experimental set-up showed that the capped αSA31 binds to MRSA specifically, and significantly (*p* < 0.05) blocks adhesion to epithelial cells. The capped αSA31 aptamer was also readily bound by preformed anti-α-gal Abs both in vivo and in vitro. The pretreatment of MRSA with αSA31, followed by the addition of monoclonal anti-α-gal IgM (mAb), blocked the phagocytosis of MRSA by mouse macrophages, indicating that mAb readily detected αSA31 on the surface of MRSA. Furthermore, the treatment of mice with αSA31 doses ranging from 300 µg/kg/day to 10,000 µg/kg/day resulted in the depletion of anti-α-gal Abs from serum. The reduction in free anti-α-gal Abs from serum indicates the circulating preformed Abs successfully detected and bound the αSA31 aptamer. This same trend of Abs reduction was also seen with 1200 µg/kg/day and 10,000 µg/kg/day of αSA31 plus vancomycin during MRSA infection, though the depletion in these cases was not statistically significant.

To clear infection, ideally, host-produced serum anti-α-gal Abs would bind to the αSA31 on the surface of a MRSA cell and activate the alternative complement pathway. The alternative complement pathway activation is tightly regulated and is a primary line of defense against infection, but many Gram-positive bacteria, especially *S. aureus*, are resistant to the lysis methods of this pathway [51,52,53]. Instead, many Gram-positive bacteria are cleared via phagocytosis and the classical complement pathway [54,55]. This interaction between the host’s own complement pathways, the αSA31 aptamer and MRSA growth was tested. Growth curves confirmed that MRSA growth in 20% serum is not affected by the presence of αSA31, indicating αSA31 has no direct anti-MRSA activity and is not sufficient to inhibit MRSA growth. However, assays measuring phagocytosis showed that significantly more MRSA were phagocytized when they were treated with αSA31 as opposed to no pre-treatment. The addition of αSA31 does not appear to activate mechanisms of the alternative complement pathway, but does seem to increase classical complement phagocytotic activity.

The in vitro characterization experiments show that αSA31 did not exhibit MRSA growth inhibition but did increase MRSA phagocytosis by macrophages, indicating a potential ability to control MRSA infection in vivo. Prior to in vivo application, however, aptamer modifications were tested for their stability and potential toxicity. Aptamer modification positively altered stability in serum, leading to the selection of a capped α-mer that was resistant to nuclease digestion for use during the rescue experiments. The in vivo application of αSA31 showed the aptamer was removed by circulating anti-α-gal Ab, but no toxicity was observed, even at 10,000 µg/kg/day. Considering the increased stability and lack of toxicity, even at high concentrations, αSA31 was applied to control MRSA infection in septic mice.

Having demonstrated that anti-α-gal Abs can bind the aptamer in vivo and that the α-mer specifically interacts with, and increases the phagocytosis of, MRSA in vitro, we hoped to show that αSA31 alone could rescue infected mice. While complement may not result in direct lysis, opsonization via anti-α-gal Abs should result in the increased phagocytosis and killing of bacteria by immune cells [56,57,58,59,60]. Mice infected with MRSA and then treated with αSA31 alone, however, were not rescued. The infectious dose of MRSA (1 × 10^9^ CFU/mouse) may have been too high to show a survival effect with an indirect treatment, like αSA31, alone, as opposed to vancomycin, which acts directly on the microbe.

While survival was not affected by treatment with αSA31 alone, the overall bacterial load in the organs was significantly affected by the presence of αSA31, especially when used in combination with vancomycin. In all organs except the spleen, mice treated with vancomycin plus αSA31 had lower CFU/mg organs than mice treated with vancomycin alone. This was particularly prominent in the lung, where the bacterial load of the vancomycin alone group was significantly higher than the aptamer–vancomycin group. High loads of MRSA in the lung can result in life-threatening pneumonia and is a common complication of nosocomial MRSA infection [61]. The reduced bacterial load in the lung seen with the vancomycin and aptamer treatment in this study presents a promising result for the reduction in sepsis-related issues, such as pneumonia, through aptamer treatment.

The treatment effect seen with αSA31 application supports the hypothesis that αSA31 tags MRSA for immune detection and assists in clearing the MRSA infection, but only as an adjuvant to vancomycin. Vancomycin is a glycopeptide antibiotic that slows the growth of MRSA through the inhibition of cell-wall synthesis. The slowed growth from vancomycin treatment may increase opportunities for the interactions between αSA31-tagged MRSA and the immune system, ultimately leading to increased phagocytosis by macrophages. This may also explain the higher concentration of MRSA CFU/mg spleen observed in the combination treatment group, as macrophages with phagocytized MRSA traffic back to the spleen [62]. The cooperation of vancomycin and αSA31 needs further research to determine if the synergistic effect could lower the minimum inhibitory dose of vancomycin, as seen in other synergistic drug studies [48,63].

There may be an even more efficient approach for signaling the immune system with α-mers than targeting the whole cell with a single aptamer. The use of multiple capped α-aptamers could lead to a higher amount of α-gal on the MRSA cell surface, as demonstrated by Cao et al., with multiple aptamers and increased fluorescence [32]. Alternatively, targeting more specific surface proteins of MRSA could improve the binding kinetics of the aptamer, possibly increasing survival in animals through improved immune activation or through more direct protein inhibition.

A known evader of the immune system, MRSA employs many evasive and anti-phagocytic strategies to go undetected by the innate immune system [64]. Several surface proteins, such as staphylococcal surface protein A (SpA), staphylococcal immunoglobulin-binding protein (Sbi), and heme-uptake proteins in the Isd family, are anti-phagocytic [52,65,66,67,68,69,70]. SpA, for example, is capable of binding the Fc portion of IgG, allowing for MRSA to cover itself in host Abs and inhibit Fc receptor-mediated phagocytosis, while Isd proteins have been implicated in the degradation of opsonic molecules [67,68]. The application of α-mers in this experiment was to reduce MRSA infection through activation of the host immune system, but another possible mechanism of action for α-mers is the interruption of such immune-evading surface proteins as SpA or Sbi. If α-mers that target these proteins could interrupt their function as well as signal the immune system, the α-mers would then have two inhibitory mechanisms against MRSA, which may result in increased survival when used alone, or lower bacterial loads synergistically, as seen with vancomycin plus αSA31.

While the rescue of MRSA-infected mice was not achieved with αSA31 alone, a synergistic effect was observed when vancomycin and αSA31 were used in combination. Though further optimization of this antibiotic–adjuvant is clearly required before it can be applied clinically, the strategy of targeted aptamer and antibiotic infections could prove extremely beneficial in the treatment of sepsis for a wide range of antibiotic-resistant bacteria.

## Figures and Tables

**Figure 1 microorganisms-11-01776-f001:**
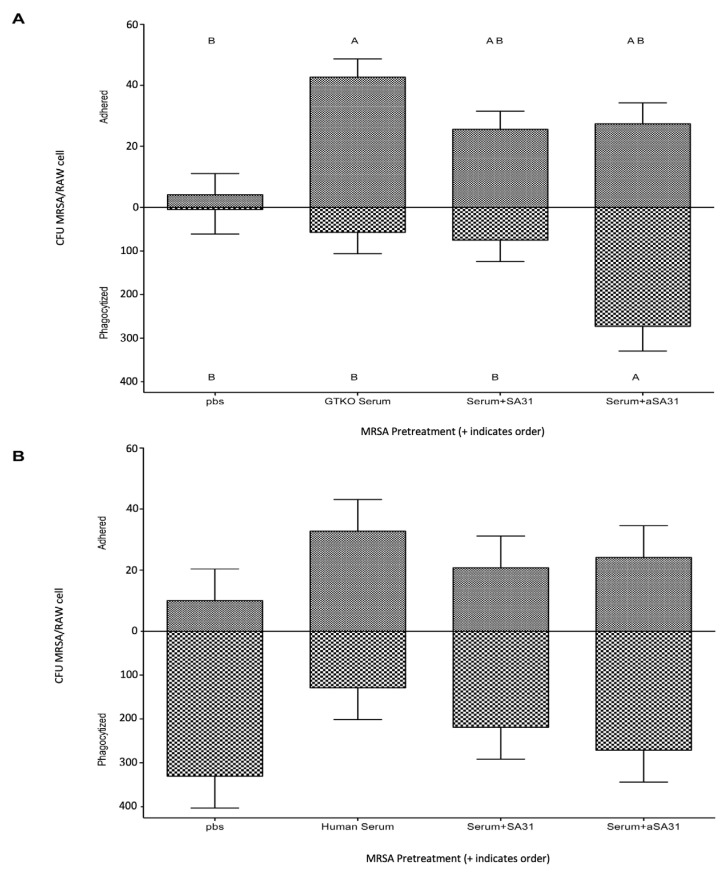
αSA31 increases the phagocytosis of MRSA in vitro. (**A**) MRSA was incubated in the presence or absence of 20% GTKO mouse serum in PBS prior to incubation with/without 150 ng/mL aptamer. Pretreated MRSA was subsequently added to RAW cells for 60 min at MOI = 100. Non-phagocytized cells were removed by washing (3x) and gentamycin treatment (100 µg/mL for 120 min). (**B**) MRSA was incubated in the presence or absence of 20% pooled human serum in PBS and then incubated with/without 150 ng/mL aptamer prior to addition to thp-1 cells at MOI = 100. Adhered and phagocytized CFU MRSA per cell was measured using qPCR. Significance was established with 1-way ANOVA and Tukey’s HSD post-test using JMP10. Bars not connected by the same letters are significantly different.

**Figure 2 microorganisms-11-01776-f002:**
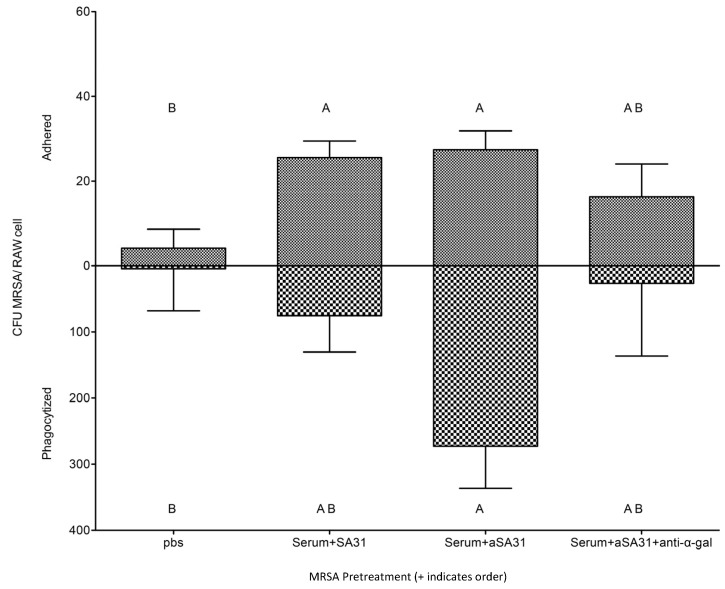
Anti-α-gal IgM interferes with αSA31 bound MRSA phagocytosis in vitro. MRSA was incubated in the presence or absence of 20% GTKO mouse serum in PBS prior to incubation with/without 150 ng/mL aptamer. Pretreated MRSA was subsequently incubated in the presence/absence of anti-α-gal IgM, M86, diluted 1:20 in PBS prior to addition to RAW cells at MOI = 100. Adhered and phagocytized CFU MRSA per cell was measured using qPCR. Significance was established with 1-way ANOVA and Tukey’s HSD post-test using JMP10. Bars not connected by the same letters are significantly different.

**Figure 3 microorganisms-11-01776-f003:**
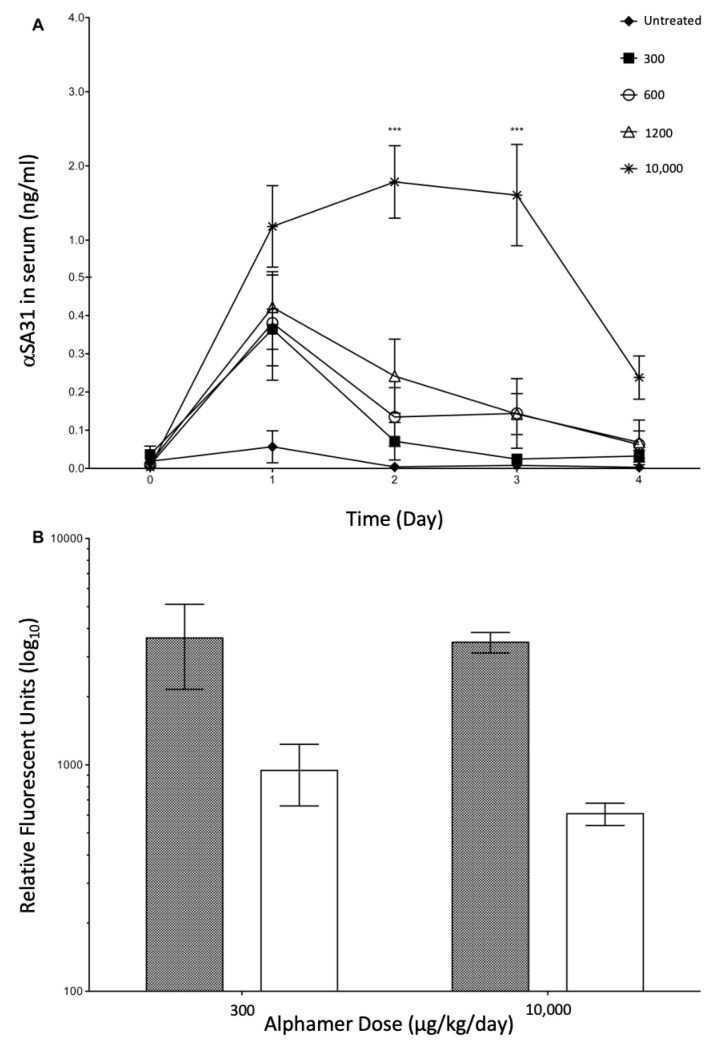
αSA31 stability and detection in uninfected mice. (**A**) In vivo stability of αSA31 in uninfected mice. Uninfected mice (n = 4) were injected twice daily with the indicated concentration of αSA31. Serum was collected prior to morning dosage. The serum concentration of αSA31 was determined by qPCR. *** Indicates a significant difference (*p* < 0.001) in serum αSA31 concentration between the 10,000 αSA31 group and all other groups on the indicated days, as determined by a 2-way repeated measures ANOVA. (**B**) Depletion of available anti-α-gal Ab after treatment with αSA31. Mice immunized against α-Gal were injected with 300 µg/kg/day or 10,000 µg/kg/day αSA31. Anti-α-gal Ab response was measured by ELISA before treatment (grey) and one day after (white) initiation of treatment using FITC labeled Ab (excitation 495 emission 535). Levels of available anti-α-gal Ab were significantly decreased following treatment (*p* < 0.0008). Statistics were from a 1-way ANOVA and a Bonferroni multiple comparisons post-test using JMP10.

**Figure 4 microorganisms-11-01776-f004:**
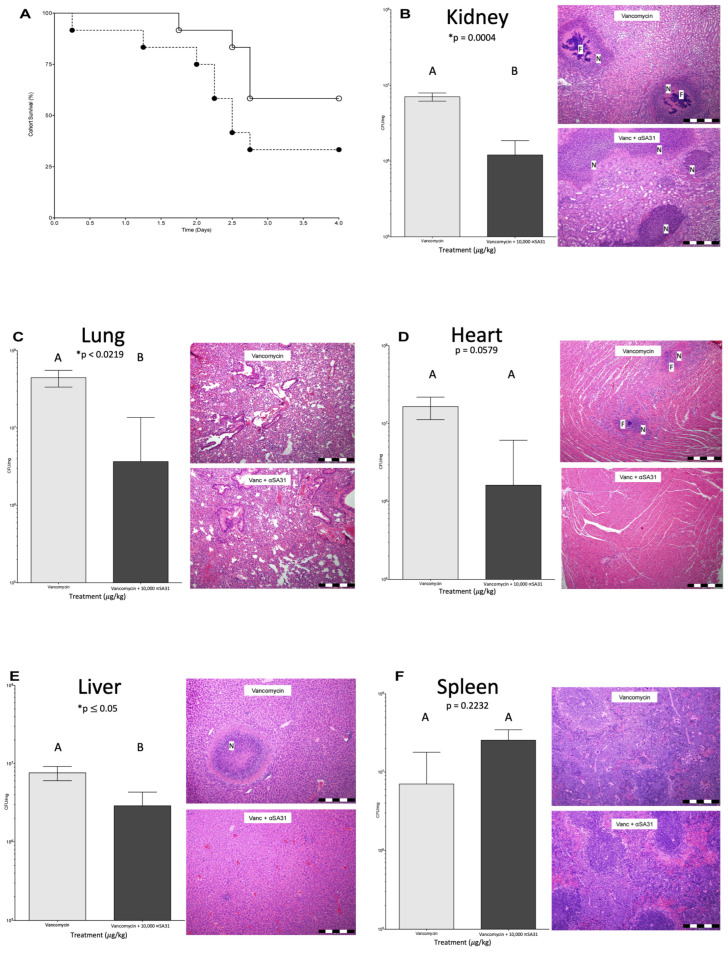
Survival and MRSA organ load of mice infected with MRSA and treated with αSA31. (**A**) mice (n = 12) were infected via tail vein IV with 10^9^ CFU MRSA and then treated with vancomycin (closed circle) and vancomycin plus αSA31 (open circle) twice daily. The (**B**) kidney, (**C**) lung, (**D**) heart, (**E**) liver, and (**F**) spleen, were collected at necropsy. Half of the organs were homogenized and lysed to determine the bacterial load in CFU/mg organ by qPCR. Significance was established using Student’s *t*-test in JMP10 for CFU/mg organ, with results segregated based on mouse survival (n = 12). Bars not connected by the same letter are significantly different. A pathologist blind to the experimental conditions performed histology. F represents a focus of infection and N represents necrosis. Markings on scale bars in tissue images represents 500 μm. * *p* indicates significant value at *p* < 0.05.

**Table 1 microorganisms-11-01776-t001:** The average half-life, in hours, of 5′-NH2-SA31-OH-3′ (SA31), 5′-α-gal-SA31-NH_2_-3′ (αSA31NH_2_), or 5′-α-gal-SA31-3C-3′ (αSA31) in the presence or absence of nucleases. * Degradation not observed after 24 h; half-life set to 10,000 h. Data with common letters are not significantly different (*p* > 0.05).

		Aptamer	
Treatment	SA31	αSA31NH_2_	αSA31
Water	69.337 ^B^	10,000 *^A^	10,000 *^A^
Endonuclease (DNase1)	1.178 ^D^	1.689 ^D^	1.233 ^D^
3′->5′ Exonuclease (Exo1)	5.014 ^C,D^	9.494 ^C,D^	4.082 ^C,D^
5′->3′ Exonuclease (T5)	1.88 ^D^	1.761 ^D^	2.099 ^D^
Human serum (homo sapiens)	3.938 ^C,D^	17.618 ^C^	5.222 ^C,D^
Mouse serum (mus musculus)	1.484 ^D^	2.467 ^D^	1.789 ^D^

## Data Availability

Data are provided in the manuscript.

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
