# Peer review of "Alpha-Gal Bound Aptamer and Vancomycin Synergistically Reduce Staphylococcus aureus Infection In Vivo"

_microorganisms, 2023, doi:10.3390/microorganisms11071776_

Round 1
Reviewer 1 Report
Dear Authors,
After reading the manuscript entitled " Alpha-gal bound aptamer and vancomycin synergistically reduce Staphylococcus aureus infection in vivo", I consider it appropriate to be published in Microorganisms in the special issue “Staphylococcal Infections”.
The subject of the manuscript matter is original and important. The Introduction section presents background information, this part is concise and well‐written. The methods present in the study are well described in detail. The main results confirm the objectives proposed by the authors. References are correlated well with the text.
Minor corrections in the manuscript text must be performed to increase its quality:
In the title – Staphylococcus aureus and in vivo should be written in italics.
line 59 is:.. Amoxicillin... should be:.. amoxicillin…
line 120 is:.. Cao et al. aptamer..., should be:... Cao et al. […] aptamer… – enter references in parentheses
Figure 3 (line 431) - the description of this figure is incomplete.
Line 456 is:.. Figure 3..., should be:… Figure 4 …
Figure 4 - the axis descriptions and the captions in the pictures are unreadable.
line 511 is:.. Cao et al. showed..., should be:... Cao et al. […] showed… – enter references in parentheses
Author Response
Reviewer 1:
In the title – Staphylococcus aureus and in vivo should be written in italics.
“Alpha-gal bound aptamer and vancomycin synergistically reduce Staphylococcus aureus infection in vivo”
line 59 is:.. Amoxicillin... should be:.. amoxicillin…
“One example is the commonly prescribed antibiotic-adjuvant mixture Augmentin, which combines the β-lactam antibiotic amoxicillin with the β-lactamase inhibitor clavulanic acid. The use of Augmentin, as opposed to amoxicillin alone, restores activity against amoxicillin-resistant bacteria and increases bacterial clearance [11-13].”
line 120 is:.. Cao et al. aptamer..., should be:... Cao et al. […] aptamer… – enter references in parentheses
“This present study builds on the study by Cao et al. [32] aptamer study and proposes an aptamer from their study, SA31, coupled to a-gal (aSA31) would rescue of a-1, 3-galactosyltransferase (-/-) knockout (GTKO) mice from induced MRSA sepsis.”
|
Figure 3 (line 431) - the description of this figure is incomplete.
Line 456 is:.. Figure 3..., should be:… Figure 4 …
Figure 4 - the axis descriptions and the captions in the pictures are unreadable.
Thank you for this comment, we have changed the axis titles to be more readable.
line 511 is:.. Cao et al. showed..., should be:... Cao et al. […] showed… – enter references in parentheses
“Previous work from Cao et al. [32] showed that SA31 modified with fluorescent probe FITC binds to S. aureus specifically, but other 5’ or 3’ alterations could impact the binding of SA31 to MRSA.”
Reviewer 2 Report
The antibacterial activity of an aptamer, namely αSA-31, a single-stranded nucleotide has been previously reported against methicillin-resistant Staphylococcus aureus (MRSA) strains. Vancomycin is a known antibiotic used for MRSA treatments. In this present study, the authors describe the combined inhibition effect of αSA-31 and vancomycin against MRSA using a mice model. They found the combination of αSA-31 and vancomycin is more efficient and provides better protection against MRSA strains. Furthermore, the authors investigate mechanistically the role of vancomycin and αSA-31 by measuring other parameters like adhesion, phagocytosis, and bacterial load into different organs of mice. This article is informative and well-written.
Minor comments:
Figure legends of Figure 4 are missing.
The labeling of the figures is small and not clear. Make it bigger to see it clearly.
Author Response
Dear reviewers,
Thank you for taking the time to read our manuscript and provide thoughtful feedback. We are glad that both Reviewers believe the manuscript needed only minor revisions. These revisions have all been made and can been found below or highlighted in the main text.
Reviewer 2:
Figure legends of Figure 4 are missing.
The labeling of the figures is small and not clear. Make it bigger to see it clearly.
Thank you for this comment. We have increased the text size in the figures to make it more readable.